# Temporal and Spatial Evaluation of Mono(2-ethylhexyl) Phthalate (MEHP) Detection in Common Bottlenose Dolphins (*Tursiops truncatus*) from Sarasota Bay, Florida, USA

Miranda K. Dziobak [1,2,*] , Brian C. Balmer [3] , Randall S. Wells [4] , Emily C. Pisarski [5] , Ed F. Wirth [5] and Leslie B. Hart [6,*]

1 Environmental and Sustainability Studies Graduate Program, College of Charleston, Charleston, SC 29424, USA
2 Environmental Health Sciences Graduate Program, University of South Carolina, Columbia, SC 29208, USA
3 Dolphin Relief and Research, Clancy, MT 59634, USA; bcbalmer1@gmail.com
4 Chicago Zoological Society's Sarasota Dolphin Research Program, c/o Mote Marine Laboratory, Sarasota, FL 34236, USA; rwells@mote.org
5 National Oceanic and Atmospheric Administration, National Ocean Service/National Centers for Coastal Ocean Science, Charleston, SC 29412, USA; emily.pisarski@noaa.gov (E.C.P.); ed.wirth@noaa.gov (E.F.W.)
6 Department of Health and Human Performance, College of Charleston, Charleston, SC 29424, USA
* Correspondence: mdziobak@email.sc.edu (M.K.D.); hartlb@cofc.edu (L.B.H.)

**Abstract:** Phthalates are endocrine-disrupting chemicals added to plastics, personal care products, cleaning solutions, and pesticides. Extensive use has led to its exposure to wildlife, including common bottlenose dolphins (*Tursiops truncatus*) from Sarasota Bay, Florida, USA; however, there are gaps in knowledge regarding whether sample timing or geographic location influence exposure. Dolphins were evaluated for temporal and spatial variability in urinary mono(2-ethylhexyl) phthalate (MEHP) detection (2010–2019). Significant fluctuations in detectable MEHP concentrations were found across the dataset. All samples from 2014 and 2015 (*n* = 12) had detectable MEHP concentrations; thus, data were classified into cohorts to explore the significance of prevalent MEHP detection ("Cohort 1" (*n* = 10; 2010–2013), "Cohort 2" (2014–2015), and "Cohort 3" (*n* = 29; 2016–2019)). Compared to Cohorts 1 and 3, Cohort 2 had higher detectable MEHP concentrations (Dunn's; *p* = 0.0065 and *p* = 0.0012, respectively) and a greater proportion of detectable MEHP concentrations (pairwise comparisons using Benjamini–Hochberg adjustments: *p* = 0.0016 and *p* = 0.0059, respectively). MEHP detection also varied across spatial scales. Dolphins with detectable MEHP concentrations had ranges primarily within enclosed embayments, while dolphins with nondetectable MEHP concentrations extended into open waters, potentially indicating geographically linked exposure risk. This study suggests that researchers and management agencies should consider a population's ranging pattern, geographic habitat characteristics, and sample timing when assessing small cetacean health in relation to contaminant exposure.

**Keywords:** phthalates; ranging pattern; photo-identification; bottlenose dolphin

## 1. Introduction

Phthalates are high-production chemicals [1] with global manufacturing exceeding 11 billion tons annually [2]. Phthalates are extremely versatile and therefore are used in a wide range of industrial products, including plastics, personal care products, packaging, insecticides, pesticides, and medical devices [3–6]. Phthalates are not chemically bound to the products they modify and can easily leach into the environment via wastewater, industrial releases, manufacturing disposal, and breakdown from products containing phthalates [1,7]. These inputs have been reported to contaminate freshwater [8,9], seawater [10], sediment [9,11], soil [12,13], the atmosphere [14,15], and the non-target biota in these matrices [16–18]. The pervasive use and migration of phthalates from consumer

products have resulted in widespread marine contamination reported in areas that range from remote (e.g., Aleutian Islands [19] and the High Arctic [20]) to urbanized (e.g., coastal areas along the Persian Gulf [21] and the Qiantang River, China [22]) locations. In fact, evidence of exposure among marine wildlife is abundant and diverse across all trophic levels including exposure to benthic species (e.g., blue mussels *Mytilus edulis* [16]; dungeness crabs, *Cancer magister* [16]; prawns [23]), intermediate consumers (e.g., white-spotted greenlings, *Hexogrammos stelleri* [16]; Dolly Vardens, *Salvelinus malma* [24]; silver scabbardfish, *Lepidopus caudatus* [25]; common roach fish, *Rutilus* [26]), and higher-level predators (e.g., Atlantic bluefin tuna, *Thunnus thynnus* [27]; harbor porpoises, *Phocoena* [28]; fin whales, *Balaenoptera physalus* [18]; Risso's dolphins, *Grampus griseus* [29]; striped dolphins, *Stenella coeruleoalba* [29]; and common bottlenose dolphins, *Tursiops truncates* [29–32]). While exact sources of marine phthalate exposure are not well-understood, marine debris may be a significant contributor. There are an estimated 5 trillion pieces of plastic floating in the ocean [33] that can release phthalates directly into the marine environment [34].

Interest in environmental phthalate contamination has increased due to their endocrine-disrupting propensity [35,36]. Much of the knowledge regarding phthalate health impacts has been informed by laboratory studies. For example, exposure to two commonly used phthalates, di(2-ethylhexyl) phthalate (DEHP) and dibutyl phthalate (DBP), have been linked to antiandrogenic activity leading to reproductive tract malformations in male rats [37]. Prenatal exposure to DBP has also been shown to significantly decrease serum testosterone levels in rats [38]. Another phthalate, diisononyl phthalate (DiNP), has been identified to disrupt sexual differentiation in European pikeperch (*Sander lucioperca* [39]). Widespread exposure reported in human epidemiological studies have linked phthalates with these reproductive impacts, as well as neurodevelopmental changes in infants and children [40–44], obesity [45], and diabetes in adolescents and adults [46]. Thus, increases in plastic marine debris and the detection of phthalates in marine mammals are concerning for wildlife health [40–44].

Sarasota Bay is a coastal, semi-enclosed lagoon system consisting of several smaller embayments, located along the central, west coast of Florida [47–49]. Water circulation throughout this bay area is generally limited and driven by minimal tidal exchange (maximum of ~1 m [50]) with the Gulf of Mexico [49]. As each of these embayments has its own set of conditions and influences, there may be variability in the type and concentration of contaminants due to these geological and environmental factors [49]. Various methodologies have been employed since 1970 to study the year-round, multi-decadal, multi-generational resident bottlenose dolphin community in Sarasota Bay, Florida, including systematic monthly and/or seasonal photo-identification surveys, radio and satellite-linked telemetry, focal animal behavioral observations, periodic health assessments, and stranding response [51]. As a result of these long-term research efforts, approximately 96% of the dolphins in the Sarasota Bay study area can be recognized using standardized photo-identification techniques [52].

Recently, studies of dolphins inhabiting Sarasota Bay, FL have revealed prevalent exposure to phthalates (~75% of sampled individuals [30–32]). Mono(2-ethylhexyl) phthalate (MEHP), the first metabolite of di(2-ethylhexyl) phthalate (DEHP), was the most frequently detected metabolite in Sarasota Bay dolphins (~55% of individuals [30]) and had a geometric mean concentration exceeding human reference populations [31]. This variation might indicate differences in exposure routes between dolphins and humans; however, spatial and temporal aspects of exposure in the Sarasota dolphin population are unknown. Previous studies have demonstrated that proximity to human development (e.g., factories and manufacturing sites [10]) can influence overall phthalate exposure [10,53]. Surrounding Florida's 14th most populous city [54], the Sarasota Bay watershed is predominantly developed with high density residential use, industrial facilities, and agricultural land operations. In fact, only 1% of the watershed is under protection (approximately 2 square kilometers), with the majority of the land use considered "urban, built up" (more than 66% [55]). Waterways located near areas with urban land use have been identified as sinks

for various environmental pollutants [56,57]. The Sarasota Bay dolphin community containing approximately 160 individuals [51] has been growing at an annual rate of 2.1% [58] and is part of the world's longest-running study of a wild dolphin population. Long-term data collected on individual animals provide a unique opportunity to evaluate potential spatial and temporal differences in phthalate exposure within this population [47,51,59,60]. Individual dolphins are routinely observed during systematic, monthly photo-identification surveys documenting their movement patterns, body condition, and activities [51], facilitating the study of temporal and spatial differences in phthalate exposure. These survey data distinguish individual dolphins using the pattern of markings (e.g., scarring, nicks, notches) found along the dorsal fin [61,62]. The trailing edge of the dorsal fin is sensitive to tearing, resulting in long-term notches unique to each individual [62–64]. Dorsal fin photo-identification data can then be paired to urine samples collected during capture–release health assessments for phthalate analyses. The combination of these sampling methodologies, which have been previously used in studies of other chemical contaminants (e.g., polychlorinated biphenyls, PCBs [65–67]; organochlorine pesticides, OCPs [65–67]; and polybrominated diphenyl ethers, PBDEs [65,66]), provides an opportunity to understand movement patterns of exposed dolphins.

The overall objective of this study was to examine temporal and spatial parameters and how they relate to the variability in MEHP detection among Sarasota Bay dolphins across a 10-year sampling period (2010–2019). Specifically, we aimed to examine the temporal trends in phthalate metabolite detection as a surrogate for exposure from dolphins sampled from 2010 through 2019 and compare ranging patterns between exposed and unexposed dolphins. Building upon previous phthalate exposure research conducted on Sarasota Bay dolphins [30,32], this study will enhance our understanding of exposure risk across spatial and temporal scales and provide a foundation to identify other populations that may be at high risk of exposure.

## 2. Materials and Methods

### 2.1. Dolphin Community and Study Location

Dolphins sampled for this study were individuals considered to be resident to Sarasota Bay ($n$ ~160 [47,51]; Figure 1). Urine samples ($n$ = 69) were collected opportunistically during catch-and-release health assessments [60] annually from 2010–2019. Dziobak et al. (2021) screened and reported concentrations for the phthalate metabolites monomethyl phthalate (MMP), monoethyl phthalate (MEP), MEHP, mono-2-ethyl-5-hydroxyhexyl phthalate (MEHHP), mono(2-ethyl-5-oxohexyl) phthalate (MEOHP), monobenzyl phthalate (MBzP), monoisobutyl phthalate (MiBP), and monobutyl phthalate (MBP). Urine is sensitive enough to detect phthalate metabolites at low concentrations and consistently yields accurate measurements [68,69]. Phthalate metabolites were screened given the potential for contamination from parent compounds in plastic sampling equipment used throughout analysis. These samples represent 51 unique individuals, as some dolphins ($n$ = 13) were sampled repeatedly; however, repeated samples from these individuals were not included in analysis. MEHP was the most frequently detected phthalate metabolite (~55% of individuals [30]) and was therefore selected as the target metabolite of this study; the remaining metabolites were not detected frequently enough to conduct statistical analyses.

### 2.2. Phthalate Exposure Assessment

Urine samples for this study (2010–2019) were collected during routine health assessments conducted under Scientific Research Permits #522-1785, #15543, and #20455 issued by the National Oceanic and Atmospheric Administration's (NOAA) National Marine Fisheries Service (NMFS). All capture and sampling methodologies for the health assessments were reviewed and approved annually by the Mote Marine Laboratory's Institutional Animal Care and Use Committee (IACUC). Standardized urine collection methods previously described [32,51] were consistent over the course of the 10-year sampling period. Analysis, quantification, quality control methods, and detection limits for urinary phthalate

metabolite screening have been previously described and reported [30,32]. These methods were based on protocols established by the Centers for Disease Control and Prevention (CDC) for human phthalate analysis (CDC, 2012). Urine samples were processed in batches along with quality assurance/quality control (QA/QC) samples (reagent blanks, field blanks, reagent spikes, matrix spikes, and SRM 3672 Organic Contaminants in Smokers' Urine [30,32]). Sample integrations were performed using Analyst software (ver 1.5).

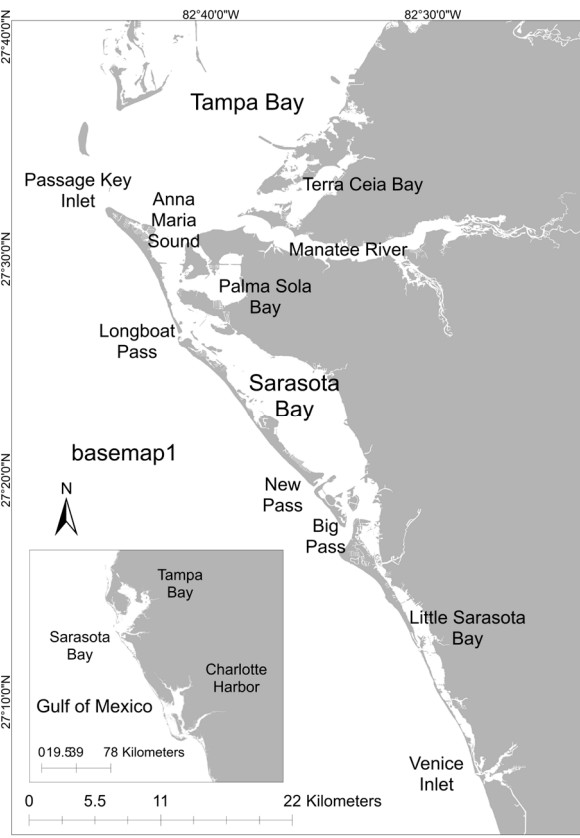

**Figure 1.** Sarasota Bay study area. The same area was utilized for photo-identification efforts and urine sampling sites.

## 2.3. Photo-Identification Records

Photo-identification of a dolphin's dorsal fin has been widely used as an identifying feature to differentiate individual dolphins [64]. The survey and sampling methodologies for photo-identification have been described elsewhere [51,70]. Small vessel surveys are commonly employed to collect photo-identification data within a given study area [51]. Standardized photo-identification surveys have been ongoing since 1980, but consistent, year-round surveys began in 1992 [51,70]. The dolphin sighting histories were obtained from the Sarasota Dolphin Research Program (SDRP) and included more than 20,000 individual sighting records.

Currently, pathways for phthalate metabolism in dolphins are largely unknown, including the temporal scale to process and excrete phthalate metabolites. As such, the selection criteria for the photographic records used for this analysis were carefully considered. Human studies have shown metabolism occurs rapidly and is followed by metabolite excretion through urine or feces [71,72]. Daily fluctuations in phthalate metabolite concentrations have been observed in human (urine) samples [73,74]; however, single spot samples have been used as a predictive metric for average phthalate concentrations over the previous 3–6 months [73,74]. Since this has yet to be studied in dolphins, sighting histories were limited to 12 months prior to urine sampling to account for the potentially rapid metabolism while still providing a conservative ranging pattern estimate for each cohort.

### 2.4. Temporal Analysis

Descriptive statistics (e.g., geometric means and 95% confidence intervals) were used to summarize and evaluate detectable MEHP concentrations by sampling year (2010–2019). For years when more than 20% of sampled dolphins had MEHP concentrations below the limit of detection (LOD; hereafter referred to as "nondetects") the mean and standard deviation were calculated using Robust Regression on Order Statistics (ROS) and Kaplan–Meier (K–M) methods [75,76]. Although they have been commonly used, substituting values (such as LOD/2) have been shown to perform poorly when used to estimate descriptive statistics with censored data [77]. All dolphins sampled during 2014 and 2015 had detectable concentrations of MEHP, something not seen in any of the other sampling years. Based on these data observations, samples from 2014 and 2015 were combined into a common category ("Cohort 2"). Remaining samples were assigned to "Cohort 1" or "Cohort 3" categories based on collection date. Only one urine sample was used from an individual for comparisons between cohorts. If a sampling event occurred during 2014 or 2015, that sample was retained, and the samples taken later were excluded. Otherwise, the most recently obtained sample was used (hereafter referred to as "retained samples," as in Table 1 [30]). Some dolphins sampled during 2014 and 2015 ($n = 4$) were sampled more than once, so patterns between repeated sampling years were also examined (Table S1).

**Table 1.** Mono(2-ethylhexyl) phthalate (MEHP) screening results for Sarasota Bay bottlenose dolphin urine samples (2010–2019).

| | Total Samples | | | Retained Samples | |
|---|---|---|---|---|---|
| Sampling Year ($n = 69$) | MEHP Detects ($n$) | % Total | Mean (s.d.) Detects and Nondetects (ng/mL) | Sampling Year [4] ($n = 51$) | Geometric Mean and 95% Confidence Interval [4] (ng/mL) |
| 2010 ($n = 9$) | 3 | 33.00% | 0.61 (1.17) [2] | 2010 ($n = 1$) | - |
| 2011 ($n = 4$) | 0 | - | - | 2011 ($n = 0$) | - |
| 2012 ($n = 10$) | 7 | 70.00% | 1.91 (0.94) [1] | 2012 ($n = 2$) | - |
| 2013 ($n = 1$) | 0 | - | - | 2013 ($n = 0$) | - |
| 2014 ($n = 8$) | 8 | 100.00% | 38.42 (24.94) [3] | 2014 ($n = 7$) | 19.35 (3.94–95.02) |
| 2015 ($n = 5$) | 5 | 100.00% | 32.12 (4.47) [3] | 2015 ($n = 5$) | 31.90 (26.90–37.82) |
| 2016 ($n = 8$) | 3 | 37.50% | 1.16 (1.10) [2] | 2016 ($n = 2$) | - |
| 2017 ($n = 10$) | 6 | 60.00% | 1.83 (1.67) [1] | 2017 ($n = 5$) | 2.03 (0.76–5.40) |
| 2018 ($n = 6$) | - | - | - | 2018 ($n = 0$) | - |
| 2019 ($n = 8$) | 7 | 87.50% | 2.06 (2.18) [1] | 2019 ($n = 7$) | 1.40 (0.48–4.05) |

[1] Calculated for all individuals including nondetects via Kaplan–Meier (Helsel, 2005; Dziobak et al., 2021). [2] Calculated for all individuals including nondetects via ROS (Helsel, 2005; Dziobak et al., 2021). [3] All values above LOD. [4] Calculated for only individuals with detectable MEHP concentrations.

The magnitude of MEHP detection was evaluated using a Kruskall–Wallis test [78], and pairwise comparisons between temporal groupings were conducted using a Dunn's test [79]. To evaluate differences in MEHP detection (yes/no), a Peto–Peto test was used to compare the proportion of non-censored observations (i.e., concentrations above detection limit) between groups [75] (NADA R package). Benjamini–Hochberg (BH) adjustment was used for post-hoc comparisons. Statistical significance was evaluated using $\alpha = 0.05$. All statistical analyses were conducted using Statistica (Version 13, Dell Inc., Round Rock, TX, USA) and R (Version 3.2, R Foundation for Statistical Computing, Vienna, Austria) software packages.

### 2.5. Spatial Analysis

Two spatial analyses were conducted: (1) a comparison between dolphins with detectable urinary MEHP concentrations (hereafter referred to as "detects") and dolphins with MEHP concentrations below the limit of detection (LOD; hereafter referred to as "nondetects") and (2) a comparison across the sampling years. (i.e., Cohort 1, Cohort 2, and Cohort 3). Spatial analyses were conducted using methods described in MacLeod (2013). Briefly, Geostatistical

Analysis and Spatial Analyst Tools (ArcGIS 10.8.1, ESRI, Redlands, CA, USA) were employed to generate kernel density estimation (KDE) accounting for barriers [80]. KDE is a non-parametric process used to estimate the probability density function for each comparison group [81]. Bandwidth, or the smoothing parameter (h), for the KDE was determined with a rule-based ad-hoc method [82]. To compare ranging patterns of detects vs nondetects, sighting histories were grouped into cumulative ranging patterns for both groups. KDE was used to determine 50% and 95% utilization distributions (UDs) to describe the core (50% UD) and total (95% UD) ranging patterns of detects and nondetects. Area of overlap between the two groups was calculated. Comparisons across sampling years were conducted using the same conditions as the temporal analysis in which groups included "Cohort 1", "Cohort 2", and "Cohort 3". Sighting histories were grouped into cumulative ranging patterns for all three temporal groups. As there are knowledge gaps regarding the timing of phthalate metabolism in dolphins, only the core (50% UD) ranging area was used to provide a conservative estimate of spatial use in relation to MEHP exposure. Areas of overlap were determined among all three groups. All spatial analyses were calculated in the Universal Transverse Mercator (UTM) Zone 17 North projection.

## 3. Results

### 3.1. Temporal Patterns of MEHP Concentrations

With the exception of 2011, 2013, and 2018, MEHP was detected in at least one sample in all years (Table 1). Detectable concentrations (i.e., concentrations > LOD) of MEHP were present in 100% of samples screened in 2014 (*n* = 8) and 2015 (*n* = 5; Table 1).

To categorize samples across the sampling period (2010–2019), MEHP means and standard deviations were plotted for each sampling year (Figure 2). Data for 2014 and 2015 stood out; therefore, a decision was made to group the rest of the sampling years relative to 2014 and 2015 as Cohort 1, 2, and 3 (as defined in the Methods section) and compare exposure among these three temporal groups (Figure 2).

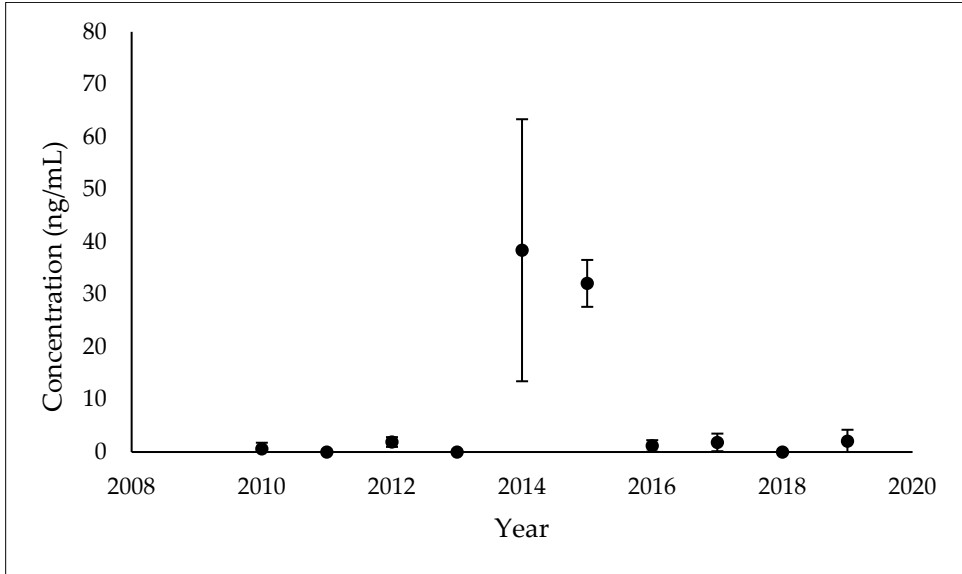

**Figure 2.** Mono(2-ethylhexyl) phthalate (MEHP) arithmetic mean concentration (dots; ng/mL) and standard deviation values (whiskers; ng/mL) for all bottlenose dolphins sampled from Sarasota Bay, Florida during 2010–2019 (values shown in Table 1). Of the 69 total samples, 13 dolphins were sampled more than once and are included in this figure to preliminarily identify natural groupings in the data.

With the exception of one dolphin, samples obtained during 2014 and 2015 appear to be elevated compared to samples obtained from the same individual during other years (Table S1). A Kruskal–Wallis test confirmed the suspected differences among temporal

groupings (Kruskal–Wallis, $\chi^2$ = 15.69, df = 2, $p$ = 0.0004). The mean MEHP concentration in Cohort 2 was significantly higher than both Cohort 1 (Dunn's, $p$ = 0.0065) and Cohort 3 (Dunn's, $p$ = 0.0012). Results from the Peto–Peto test revealed a difference in the likelihood of MEHP detection, as indicated by the proportion of concentrations above the limit of detection ($\chi^2$ = 55.2; df = 2, $p \leq$ 0.0001). Pairwise comparisons with BH adjustment between Cohort 1, Cohort 2, and Cohort 3 confirmed a significant difference in the proportion of detects between all temporal groupings (Table 2).

**Table 2.** Mono(2-ethylhexyl) phthalate (MEHP) mean and standard deviation values for temporal groups: (1) Cohort 1 (i.e., samples collected 2010–2013); (2) Cohort 2 (i.e., samples collected 2014–2015); (3) Cohort 3 (i.e., samples collected 2016–2019). Since the Peto–Peto $p$ < 0.05, pairwise multiple comparison tests were conducted between groups with Benjamini–Hochberg (BH) adjustment.

| Temporal Group | Mean and Standard Deviation (ng/mL) | Percent > LOD (95% CI) | Pairwise Comparisons of Percent Detect between Temporal Groups | |
|---|---|---|---|---|
| | | | Cohort 2 | Cohort 3 |
| Cohort 1 ($n$ = 10) | 0.43 (0.79) [1] | 30.00 (1.60–58.40) | $p$ = 0.0016 | $p$ = 0.0059 |
| Cohort 2 ($n$ = 12) | 33.71 (18.43) [2] | 100.00 (-) | - | $p$ = 0.0059 |
| Cohort 3 ($n$ = 29) | 1.24 (1.69) [1] | 51.72 (23.45–54.86) | - | - |

[1] Determined by Robust ROS. [2] All values above LOD.

### 3.2. Spatial Patterns of MEHP Concentrations

To determine whether spatial use varied between dolphins with and without detectable concentrations of MEHP, dolphin ranging patterns were compared throughout the Sarasota Bay study area (Figure 3). Detects had a larger total (95% UD) ranging pattern area (108.79 km$^2$; Table 3) than nondetects (79.17 km$^2$; Table 3). The total area associated with detects included Anna Maria Sound, Palma Sola Bay, and part of Sarasota Bay, as well as between New Pass and Venice Inlet (Figure 3). This area also included the Gulf of Mexico near Longboat Pass (Figure 3). Similarly, the total area associated with nondetects included parts of Anna Maria Sound, Palma Sola Bay, Sarasota Bay, and New Pass, as well as a small area near Venice Inlet (Figure 3). The nondetects' ranging pattern also extended into the Gulf of Mexico near Longboat Pass, but it was not as prevalent as the detects' ranging pattern (Figure 3). The ranging pattern associated with detects had a smaller core (50% UD) area (17.77 km$^2$; Table 3) than nondetects (19.67 km$^2$; Table 3). Detect core (50%) UDs were found in and around Palma Sola Bay, as well as around New Pass. Nondetect core (50%) UDs were not found south of New Pass and mainly included Palma Sola Bay and Anna Maria Sound. Total (95% UD) area had a higher degree of overlap between detects' and nondetects' ranging patterns (56% and 77%, respectively) than core (50% UD) areas (45% and 41%, respectively; Table 3). Generally, the total ranging area overlap encompassed Anna Maria Sound and Palma Sola Bay and spanned south to the northern portion of Sarasota Bay (Figure 3). A smaller area of overlap was found between New Pass and Big Pass (Figure 3). Core (50% UD) area of overlap occurred between Palma Sola Bay and Longboat Pass and a small region near New Pass (Figure 3).

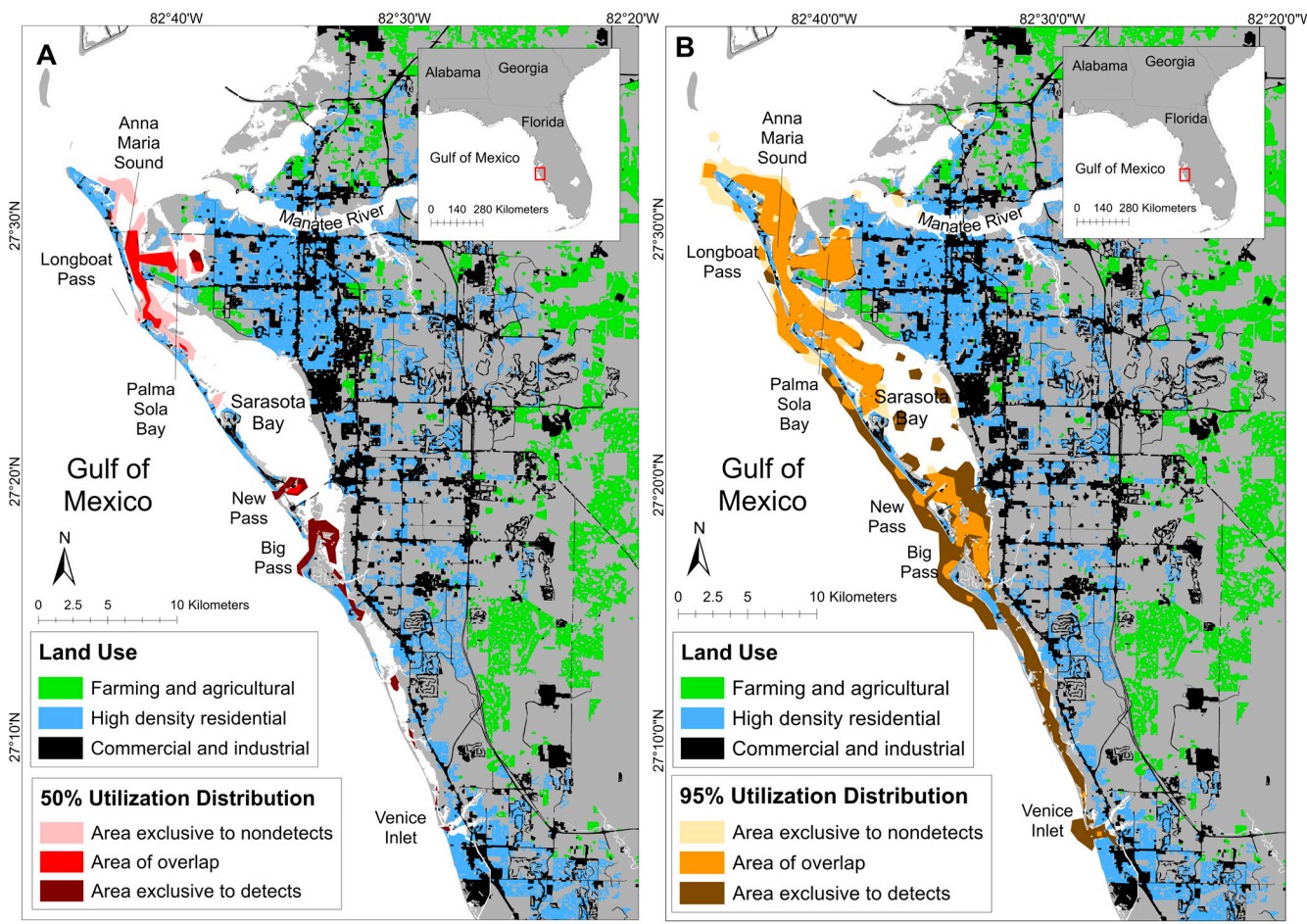

**Figure 3.** Spatial use of bottlenose dolphins in the Sarasota Bay study area (2010–2019) with detectable (detects) and nondetectable (nondetects) mono(2-ethylhexyl) phthalate (MEHP) concentrations. Photo-identification sighting histories for detect and nondetect individuals were grouped into cumulative ranging patterns with (**A**) core (50% utilization distributions (UDs)) and (**B**) total (95% UDs) areas calculated. Study area indicated by the red box. Land use layer obtained from the Southwest Florida Water Management District geospatial open data portal (2021).

**Table 3.** The 50% and 95% utilization distributions (UDs) and percentage overlap for dolphins in the Sarasota Bay study area with detectable (*n* = 30 individuals; *n* = 2040 sightings) and nondetectable (*n* = 21 individuals; *n* = 2120 sightings) mono(2-ethylhexyl) phthalate (MEHP) concentrations.

| Detection Status | Total Area (km²; 95% UD) | Total Area: % of Ranging Pattern Overlap (95% UD) | Core Area (km²; 50% UD) | Core Area: % of Ranging Pattern Overlap (50% UD) |
|---|---|---|---|---|
| Detect | 108.79 | 55.78% | 17.77 | 45.00% |
| Nondetect | 79.17 | 76.65% | 19.67 | 40.66% |
| Overlap | 60.69 | - | 8.00 | - |

Given the variation in both magnitude and proportion of MEHP detection identified in the temporal analysis, ranging patterns for each temporal grouping (Cohort 1, Cohort 2, and Cohort 3) were generated to geographically compare detection patterns across the study period (Figure 4). Cohort 1 detects had a smaller core (50% UD) area (4.89 km²; Table 4) than Cohort 1 nondetects (6.50 km²; Table 4). Cohort 3 detects had a larger core (50% UD) area (12.70 km²; Table 4) than Cohort 3 nondetects (5.26 km²; Table 4). All core (50% UD) areas (regardless of detection status or temporal grouping) included Anna

Maria Sound and parts of Palma Sola Bay (Figure 4). Cohort 1 detects' areas extended just south of Longboat Pass, and Cohort 1 nondetects' UDs ranging patterns extended as far south as Big Pass (Figure 4). In addition to Anna Maria Sound and Palma Sola Bay, Cohort 2 dolphins also utilized an area near New Pass (Figure 4). Cohort 3 nondetects' ranging patterns extended the farthest south and included areas between New Pass and Venice Inlet (Figure 4). It is unclear why Cohort 2 individuals seemed to congregate in the Anna Maria Sound and Palma Sola Bay area. The specific spatial distribution could be due to dolphin behavior or life history or related to the nature of opportunistic sampling events. The spatial distribution could also be related to MEHP detection; however, a number of factors were investigated that could be related to elevated MEHP detection (e.g., rainfall, salinity, severe weather events, and sewage spills), and none were found relevant to the sampling area or time period.

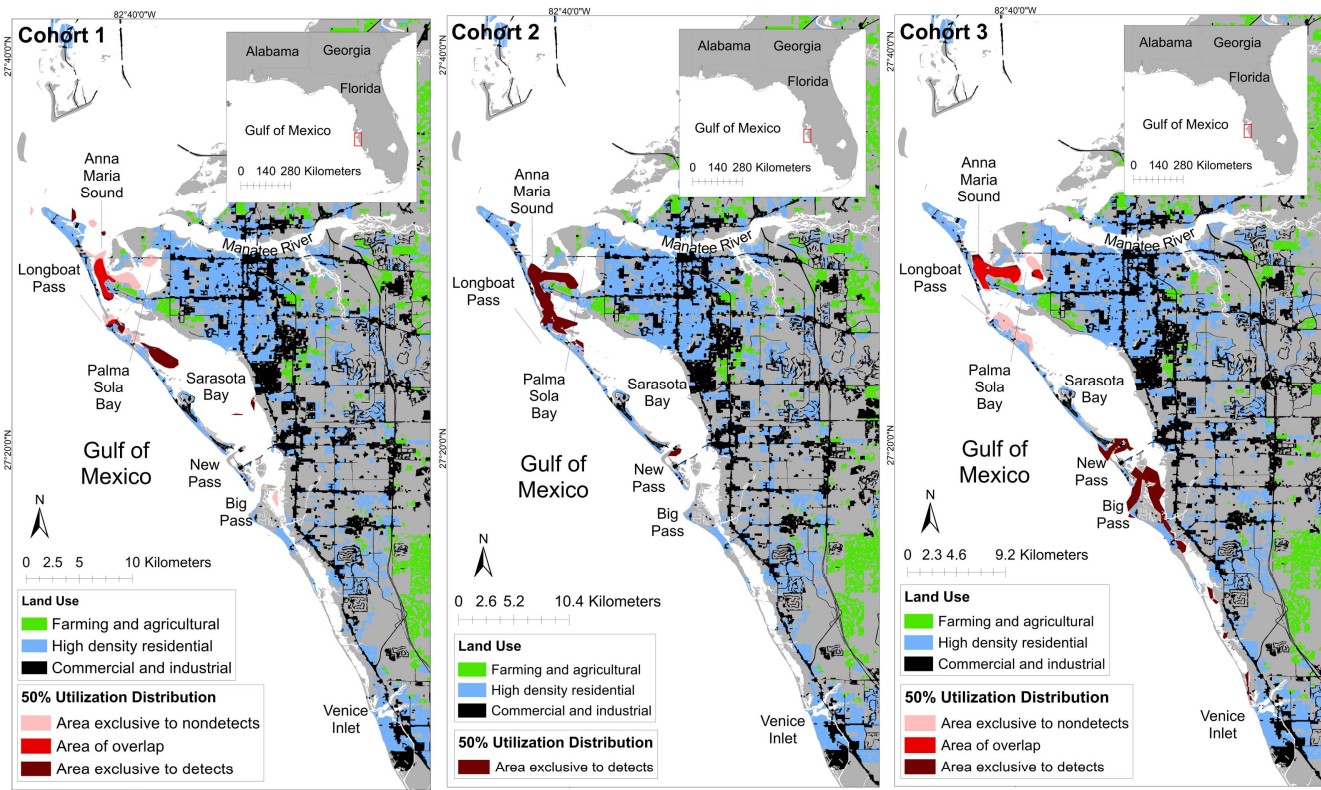

**Figure 4.** Spatial use of bottlenose dolphins in the Sarasota Bay study area (2010–2019) with detectable (detect) and nondetectable (nondetect) mono(2-ethylhexyl) phthalate (MEHP) concentrations. Photo-identification sighting histories for detect and nondetect individuals were grouped into cumulative ranging patterns with core (50% utilization distributions (UDs)) areas calculated for each temporal grouping (Cohort 1, Cohort 2, or Cohort 3). Study area indicated by the red box. Land use layer obtained from the Southwest Florida Water Management District geospatial open data portal (2021).

**Table 4.** The 50% utilization distributions (UDs) and percentage overlap for dolphins with detectable MEHP concentrations in the Sarasota Bay study area in Cohort 1 (detect: *n* = 3 individuals; *n* = 131 sightings; nondetect: *n* = 7 individuals; *n* = 171 sightings), Cohort 2 (detect: *n* = 12 individuals; *n* = 453 sightings), and Cohort 3 (detect: *n* = 14 individuals; *n* = 1502 sightings; nondetect: *n* = 15 individuals; *n* = 527 individuals) groupings.

| Temporal Grouping | Detection Status | Core Area (km²; 50% UD) | Core Area: % of Ranging Pattern Overlap (50% UD) |
|---|---|---|---|
| Cohort 1 | Detect | 4.89 | 62.64% |
| | Nondetect | 6.50 | 69.03% |
| | Overlap | 2.92 | - |
| Cohort 2 | Detect | 11.09 | 100% |
| | Nondetect | N/A | N/A |
| | Overlap | N/A | N/A |
| Cohort 3 | Detect | 12.70 | 69.03 |
| | Nondetect | 5.26 | 48.01 |
| | Overlap | 5.70 | - |

## 4. Discussion

### 4.1. Overall Findings

This study is the first spatial and temporal evaluation of phthalate metabolite detection in a dolphin population. The purpose of this research was to understand whether temporal or spatial parameters could provide insight into the variability of MEHP detection among Sarasota Bay dolphins across a 10-year sampling period (2010–2019). Ongoing studies have investigated how demographics influence phthalate concentrations in dolphins, and based upon these results, sex and age class do not appear to be dependent factors [30]. Findings from the current study identified significant differences in MEHP detection across time. In particular, 2014 and 2015 were identified as years of interest due to the high proportion of the population with detectable MEHP (100%) and high detectable MEHP concentrations (mean = 33.71 ng/mL; standard deviation = 18.43 ng/mL). Standardized urine collection methods were consistent over the 10-year sampling period, and samples were analyzed in randomly sorted batches, making it unlikely that there was any error associated with sample collection, extraction, and analysis methodologies. Single samples of human urine are thought to be indicative of average concentrations over a 3–6-month period [73,74]. If the metabolism of phthalates is similar in bottlenose dolphins, the MEHP detects as reported in this population would represent concentrations for up to 6 months prior to sample collection. Human studies have demonstrated diurnal fluctuations in phthalate metabolite concentrations in which MEHP can fluctuate 2 ng/mL within 24 h [68,83,84]. All dolphins were sampled randomly throughout daylight hours; 31 dolphins with detectable urinary MEHP concentrations were sampled in the morning (~0800–1200 EDT), and 8 were sampled in the afternoon (~1200–1600 EDT). Although it is currently unclear if dolphins have similar diurnal fluctuations in phthalate metabolite concentrations, the timing of sampling likely did not account for the variation of MEHP concentrations observed in 2014 and 2015 dolphin urine.

### 4.2. Spatial Findings

On a spatial scale, there was some evidence that habitat usage differed between individuals with detectable levels of MEHP versus those with no detectable levels of MEHP (Figure 3). The northern part of the study area, including Anna Maria Sound, Palma Sola Bay, Longboat Pass, and the north part of Sarasota Bay, did not appear to be dominated by either group. Rather, UDs for both detect and nondetect groups were found overlapping in these regions (Figure 4). Residential and agricultural activities predominantly occur in the northern portion of the study area, and by 2019, agriculture accounted for more than 1800 acres, or 3%, of land use within the watershed [55,85]. The southern portion of the

study area (New Pass to Venice Inlet) was used almost exclusively by dolphins identified with the detect group. Land use in the southern part of the watershed is primarily industrial and residential as opposed to more agricultural areas in the north [85]. Upland industrial and residential land use may further contribute to phthalate pollution through non-point-source runoff as several creeks in these areas flow into the southern portion of Sarasota Bay. Of particular concern is Phillippi Creek, which drains much of the Sarasota region (urban, suburban, agricultural), including areas with numerous septic tanks. The closure of Midnight Pass in 1983 altered water circulation through the embayments south of Big Pass [49]. As a result, water from Phillippi Creek and receiving bay waters cannot be readily flushed from this area, likely trapping and slowing the movement of contaminants.

It would be expected that industrial land use would lead to more instances of phthalate metabolite detection, as well as higher detected concentrations. In China, for example, Guangzhou is one of the largest industrial centers and one of the fastest-expanding cities in the country. Elevated phthalate levels have been found in soil sampled from this urban area, and these concentrations have been attributed to the intensity of industrial and commercial activities [86]. Conversely, phthalate concentrations from agricultural soil around the city were reported to be significantly lower than their urban counterparts [86]. Although Sarasota's industrial growth and development is not nearly as extreme as Guangzhou, the differences in MEHP concentrations detected in dolphins that reside in northern and southern sections of the study area suggest that the spatial extent of dolphin ranging patterns and surrounding land use could influence MEHP concentrations.

Spatial use throughout the study area was evaluated using the temporal groupings to determine whether detect and nondetect ranging patterns were consistent across the 10-year sampling period. If exposure is linked to habitat use, Cohorts 1 and 3 faced similar DEHP exposure risk as demonstrated by overlapping ranging patterns between detect and nondetect core (50%) UDs (Figure 4). Interestingly, habitat usage in the south appeared to shift in terms of MEHP detection status between Cohort 1 and Cohort 3 sampling years. Prior to 2014, only nondetect core (50%) UDs were identified in the southern part of the study area. This might indicate Cohort 1 dolphins predominantly residing in the south experienced lower DEHP exposure risk than those primarily inhabiting the northern part of the study area. However, there were discrepancies in sample sizes that could account for missing detects in the south; Cohort 1 had the lowest number of sampled dolphins for both detect and nondetect ranging patterns (Table 4), so dolphins residing in the south with detectable MEHP concentrations may not have been sampled. Conversely, Cohort 3 core (50%) UDs south of New Pass belonged exclusively to detects, indicating possible temporal influences regarding exposure risk. Further, in Cohort 2, all sampled dolphins had detectable urinary MEHP concentrations. These findings, paired with the significantly higher proportion and magnitude of MEHP detection in Cohort 2, suggest potential acute DEHP exposure. Acute DEHP contamination in 2014 and 2015 is supported by nondetects sampled again during the Cohort 3 temporal grouping. An acute exposure event is also supported by dolphins with repeated samplings; aside from one dolphin, samples obtained during 2014 or 2015 had higher detectable MEHP concentrations than other sampling years from the same individual. While, to our knowledge, there is no direct evidence of acute DEHP contamination in Sarasota Bay, short-term increased exposure to other organic pollutants, such as poly- and per-fluoroalkyl substances, have been associated with urban stormwater and runoff discharge [87]. Future studies should continue monitoring geographic trends in phthalate metabolite detection (particularly MEHP) and expand the sample timeframe to confirm whether acute contamination is likely to have occurred. Further, monitoring environmental samples (such as water or sediment) will also enhance understanding of marine phthalate contamination.

Acute environmental DEHP contamination has been linked with land use and urbanization. For example, during a flooding event in the River Seine in Paris, France, detectable DEHP concentrations sharply increased (96 to 1123 ng/L) in sampled river water, attributed to the interaction between flood and stormwaters and land-based sources

of DEHP (e.g., sewage systems [88]). Additional elevated DEHP concentrations that corresponded with precipitation were observed, suggesting that non-point-source runoff could influence DEHP input in an urban area [88]. In Sarasota, FL, urban areas dominate the study area's coastline, making up more than half of the watershed (66% [89]). Stormwater and urban runoff are important vectors that facilitate contaminant transportation to marine environments [90]; as water flows through areas with impervious surfaces, excess nutrients and pollutants are collected and eventually discharged into the ocean, thus exacerbating marine chemical contamination. Moreover, industrial and municipal waste disposal are the most common mechanisms of DEHP entry into the coastal environment [3]. Chen et al. (2013), in their study of the Kaohsiung Harbor, found a spatial relationship in DEHP detection in which the highest concentrations originated near the mouths of rivers discharging into the harbor. They postulated the DEHP source likely originated from upstream municipal and industrial wastewater discharges [91]. Sarasota uses an urban reclaimed water transmission system, so this mechanism of exposure may not account for continuous MEHP detection [92]. However, a failing or overwhelmed system could spill and contribute to acute DEHP pollution. For example, since 2013, the Bee Ridge Wastewater Reclamation Facility has discharged more than 800 million gallons of reuse water [93,94]. Among other freshwater creeks, Phillippi Creek has reportedly been impacted by pollutants from these spills [94]. Phillippi Creek drains into the southern area of Sarasota Bay where the largest area exclusive to dolphins with detectable MEHP concentrations was found. DEHP has been found abundantly in raw sewage samples [95], so spills such as those observed at the Bee Ridge facility may contribute to dolphin exposure. It is important to understand how urban land use impacts environmental phthalate pollution as this can directly affect local fauna. For instance, harbor porpoises sampled from the Norwegian coast were found to have phthalate metabolite concentrations seemingly dependent on human activities; the lowest detectable concentrations were found near the least-populated coastal areas, while high concentrations were significantly associated with human population sizes [28].

Finally, plastic and microplastic pollution are likely to influence marine phthalate contamination. For example, a significant correlation between microplastic abundance and phthalate ester concentration has been observed in freshwater sampled from the Lesser Himalayas [96]. Phthalates are not chemically bound to plastics and release into the environment when the plastics degrade [34]. High-density urban developments are regarded as important sources of microplastics and contribute to pollution via tire wear, industrial activities, and household laundry [97,98]. While systematic studies of microplastic pollution have not been conducted in Sarasota Bay, a previous study estimated as many as 4 billion plastic particles floating in nearby Tampa Bay [99]. As a result, Sarasota Bay may be expected to exhibit similar levels of microplastic contamination. Additionally, Sarasota Bay receives inputs from several freshwater creeks, which is concerning as microplastics are abundant in urban creeks [100]. For example, Phillippi Creek drains more than 140 square kilometers of highly developed agricultural, residential, and commercial land, which could facilitate the transport of microplastics from inland sources to the study area [49]. Further, areas where freshwater and saltwater meet impact debris movement as the density of water increases toward higher-saline waters, and these mixing zones can therefore serve as a microplastic sink [101]. It would therefore be expected that the southern portion of the study area would be more significantly impacted by microplastic pollution. Results from this study indicate the southern portion of the study area is used predominantly by dolphins with detectable MEHP concentrations, thus supporting microplastics as a potential exposure source.

Since 1986, the U.S. has produced 100 million to 250 million pounds of DEHP per year [102]. Contemporary usage of DEHP appears to have declined, as evidenced by the National Health and Nutrition Examination Survey (NHANES) reporting little to no detectable MEHP concentrations in human urine after 2012 [103]. Decreases in human detection of MEHP [103] may indicate the effectiveness of current policies regulating DEHP. Policy regarding general phthalate use and production is limited and is dictated by the Consumer Product Safety Improvement Act of 2008

from the Consumer Product Safety Commission (CPSC [104]). This act, which initially placed an interim ban on DEHP used in concentrations exceeding 0.1% in children's toys and childcare articles, was permanently enacted in 2018 [104,105]. Since implementation of these regulations corresponded with decreased human MEHP detection, results from this study suggest sources and pathways of DEHP exposure must be different for dolphins than humans. Were sources the same, it might be expected that significant changes in detection observed for one group would be paralleled in the other. Instead, Sarasota Bay dolphins have demonstrated significantly higher urinary MEHP concentrations than human reference populations [31]. Other aquatic fauna, such as American alligators (*Alligator mississippiensis*), have also exhibited elevated urinary MEHP concentrations, further indicating alternative contamination routes not observed in humans [17]. To mitigate harmful environmental phthalate exposure, additional regulation measures may need to be considered.

### 4.3. Strengths and Limitations

This study was the first to evaluate MEHP detection in bottlenose dolphins across a long-term (10-year) temporal period and a fine-scale spatial extent. Previous studies of geographic variation in phthalate detection extended across larger geographic regions (e.g., between cities rather than a single watershed) and were conducted among other species: humans [106,107], harbor porpoises [28], baleen whales (*Balaenoptera physalus* [108]), sheep [109], seabirds [19], and benthic organisms [110]. Aside from human studies, this study also represents the longest temporal study of MEHP concentrations in any mammalian species, allowing insights into phthalate detection over time. We were able to investigate phthalate exposure in dolphins using a decade of urine samples and incorporated ranging patterns using individual sighting histories that were collected across systematic surveys—a first for dolphins and possibly any marine mammal. This study relied on well-established CDC analytical methods to screen for urinary phthalate metabolites. Urine is considered an ideal matrix as it consistently yields accurate measurements and allows detection of low phthalate metabolite concentrations [68,69,111,112]. This study also relied upon urinary metabolites because sampling equipment can contaminate urine with parent phthalate compounds during analysis [69,111–113]. Long-term photo-identification efforts provided the opportunity to identify individual ranging patterns. These methods enabled the comparison of spatial distribution between exposed and unexposed dolphins overall, as well as among temporal cohorts. Spatial analysis was not based on collection site; rather, photo-identification records enabled sighting data collected for a full year prior to urine collection to generate ranging patterns. By using a year of sighting data, we were able to mitigate any distribution bias introduced by urine collection efforts. Further, ranging patterns were estimated using KDE, a method that resists outliers [80] and facilitates the identification and comparisons of core and total areas of usage.

This study utilized dolphin urine from opportunistic samplings during health assessment projects, so we were unable to control and standardize when each urine sample was taken. Further, our sampling was not completely randomized. While dolphins may be randomly encountered, some individuals may have been excluded during health assessments due to a number of factors, including age, behavior, or specific goals of funded projects. However, post-hoc analysis methods have been widely used for other chemical analyses, and methods used for this study were based on previously determined best practices [65,80,82]. This analysis was also limited by the fact that the threshold for phthalate metabolite concentrations associated with a health response in dolphins is currently unknown. Further, samples containing urinary phthalate metabolite concentrations below the LOD may not be zero. Phthalate metabolism is not understood in dolphins, so it is unknown whether measured MEHP concentrations below the LOD may be associated with potentially harmful DEHP exposure. We were also only able to collect single spot samples of urine for most dolphins. Human studies have shown single spot samples to be representative of 3–6-month average phthalate metabolite concentration; however, this has not been evaluated in dolphins [73,74]. Bandwidth selection is a critical aspect in gener-

ating accurate KDEs, as inappropriate bandwidth values can over- or under-estimate the KDE [114]. Small sample sizes also have the potential to influence biases [82]. We sought to mitigate these effects by evaluating the group ranging patterns rather than individuals, as well as by employing ad-hoc bandwidths that tend to be more resistant to sample size effects [82]. Detection status may have also influenced distribution throughout the study area, and while we are unable to speculate about whether a causal relationship exists, it should not be discounted.

## 5. Conclusions

During the 2010–2019 sampling period, samples during 2014 and 2015 had a higher proportion of MEHP detects and higher concentrations of detected MEHP than all other years. Although there are policies implemented to limit DEHP contact with humans, this study provides evidence of continued environmental contamination; therefore, further mitigation may need to be considered. This study also identified potential geographic variation in MEHP detection, indicating dolphins frequenting certain areas may be at increased risk of DEHP exposure. Widespread environmental DEHP contamination is concerning due to the risk of endocrine disruption and reproductive impairment observed in human epidemiological studies; however, the degree of impairment is currently unknown for dolphins. Future studies should evaluate phthalate exposure in relation to known health markers to determine if dolphin health may be impacted.

**Supplementary Materials:** The following supporting information can be downloaded at: https://www.mdpi.com/article/10.3390/oceans3030017/s1, Table S1: Urinary mono-(2-ethylhexyl) phthalate (MEHP) concentrations from Sarasota Bay common bottlenose dolphins (*Tursiops truncatus*) sampled more than once and those for which at least one sampling event occurred in 2014 or 2015.

**Author Contributions:** Conceptualization, M.K.D. and L.B.H.; methodology, B.C.B. and L.B.H.; validation, M.K.D. and E.C.P.; formal analysis, M.K.D.; investigation, M.K.D., R.S.W. and L.B.H.; resources, E.F.W., E.C.P. and R.S.W.; data curation, E.F.W. and R.S.W., writing—original draft preparation, M.K.D. and L.B.H.; writing—review and editing, E.F.W., E.C.P., B.C.B., R.S.W. and L.B.H.; visualization, M.K.D., B.C.B. and L.B.H.; supervision, E.F.W., R.S.W., L.B.H. and B.C.B.; project administration, R.S.W. and L.B.H.; funding acquisition, M.K.D., L.B.H. and R.S.W. All authors have read and agreed to the published version of the manuscript.

**Funding:** Funding for this work was provided by an anonymous donor and internal grants from the College of Charleston's Master of Environmental and Sustainability Studies Student Association; School of Humanities and Social Sciences; Graduate School Office; School of Education, Health, and Human Performance; and the Department of Health and Human Performance.

**Institutional Review Board Statement:** Urine samples for this study (2010–2019) were collected during routine, periodic, catch-and-release health assessments conducted under Scientific Research Permits #522-1785, #15543, and #20455 from the National Oceanic and Atmospheric Administration's (NOAA) National Marine Fisheries Service (NMFS). All catch-and-release and sampling methodologies for the health assessments were reviewed and approved annually by Mote Marine Laboratory's Institutional Animal Care and Use Committee (IACUC).

**Data Availability Statement:** Restrictions apply to the availability of these data. Dolphin location data were obtained from The Sarasota Dolphin Research Program (SDRP) and are available from the authors with the permission of SDRP. The phthalate data used for this study can be accessed through the 4TU.ResearchData, the international data repository for science, engineering, and design found at https://doi.org/10.4121/14455782 (accessed on 26 January 2021). Land use data within the Sarasota Bay coastline were obtained from the Southwest Florida Water Management District Geospatial Open Data Portal, https://data-swfwmd.opendata.arcgis.com/search?groupIds=880fc95697ce45c3a8b078bb752faf40 (accessed on 26 January 2021).

**Acknowledgments:** We thank Wayne McFee, James Daugomah, David Whitall, and journal peer-reviewers for their assistance in enhancing the manuscript. Dolphin samples were obtained through health assessments supported primarily by Dolphin Quest, Inc. (Middleburg, VA, USA). We are

grateful to the staff, collaborators, and volunteers of the Sarasota Dolphin Research Program for ensuring the safe capture, sampling, and release of the dolphins.

**Conflicts of Interest:** The authors declare no conflict of interest.

**Disclaimer:** The scientific results and conclusions, as well as any views or opinions expressed herein, are those of the author(s) and do not necessarily reflect those of NOAA or the Department of Commerce.

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
