# Peer review of "Temporal and Spatial Evaluation of Mono(2-ethylhexyl) Phthalate (MEHP) Detection in Common Bottlenose Dolphins (Tursiops truncatus) from Sarasota Bay, Florida, USA"

_2673-1924, doi:10.3390/oceans3030017_

Round 1
Reviewer 1 Report
This paper is innovative in pairing phthalate metabolite detections with the range of a species to consider where the points of pollution may exist within the ecosystem. I really enjoyed it. I'm attaching a pdf of the paper, I highlighted the spaces and comments are included with the highlights (sometimes the little post-it icon that accompanies the highlight is tiny, so I wanted to make sure you keep an eye out for the comment!). Overall the paper is well-done, but there are a handful of revisions that can be helpful for the reader. More details on the pdf but overall:
- the justification for looking at MEHP in this particular population of dolphins is not extremely clear, why MEHP specifically (and not the other metabolites you screened for, as described in the methods) and why these dolphins specifically? I realize they have been studied for a very long-time, and the photo-ID is super useful to look at the spatial distribution of exposure, but is there anything else? Is there concern over their decline? Are they helpful environmental sentinels for this system? Etc, it would be good to dive into that a little more
- the distinction between values of MEHP that are <LOD and non-detect got really confusing, especially since in the last section about study limitations you do mention that <LOD does not imply 0 but you still call <LOD non-detects (which implies 0). This was a bit confusing for me, as well as the methods in which values were retained. A bit more clarity around that would be super helpful in interpreting the results (especially on table 1, I found it confusing)
- the mapping was very cool, I really liked it, just had some comments about justification for looking at a 12 month period vs something else, you sort of address it in the last section, but would be helpful to address in the methods

Reviewer 2 Report
I really enjoyed this well written paper. I was interested to read it because there is little literature on phthalate exposure and risk in marine mammals. It is important to gain some perspective on how significant a threat plasticizers are for marine mammals, particularly those feeding higher up the food chain in which plastic ingestion is likely to be less common and incidental than those that forage at lower trophic levels (and thus exposure to plasticizers may be dismissed as a significant risk). The geographical and temporal aspect of exposure is also poorly understood and this paper capitalizes on the availability of a unique long term dataset to present this information, which is difficult to obtain for hard to access but vulnerable top predators.
The introduction provides a comprehensive yet succinct overview of the phthalates and the risks they present. The methods are clear and concise, the results clearly presented and thoroughly discussed with limitations clearly articulated and conclusions that fairly summarize what can be inferred from the data.
Abstract:
Line 15: ‘their’ needed before ‘detection’.
Without adding much extra text the following points could be made clearer:
Line 16: Before you say what was done it would be helpful to say what the knowledge gap is that this paper addresses.
Line 20: does this mean you did the analysis using presence/ absence?
Line 25: how did you know what the ranges were? How was this assessed?
Line 29: does it suggest that animals that use some areas are more at risk of these chemicals/ that risk increases when animals move into specific inshore areas/ embayments? Or does small sample size and interannual confounds prevent you from being able to say anything so definitive?
Introduction:
Line 33: ‘production’ used twice in the same sentence.
Methods:
Line 104-119: Belongs better in the introduction for context on both the area and the animals studied
Line 122: it would be good to clearly make reference to the fact that efforts were made to ensure that the sampling and storage of the samples were phthalate free given there is a risk that storage in plastic and that use of plasticware in the sampling protocol can contaminate the samples. Ref 30 describes this process but it would be good to mention that blanks were taken to ensure positives are true positives, even though plastic is used in the collection and storage of the samples.
Line 166: The way this is written it suggests you pooled the sightings to get at ranging pattern rather than doing that per individual? The reasons for this could be more clearly stated here
Results:
Table 1 and fig 2 and 3 are especially helpful. Fig 3 is really needed to help make sense of the text about the geographical differences.
Line 301-302: how were these potential explanatory variables investigated?
Discussion:
Line 328 -330: I like that you have included this point to show that batching of runs cannot have generated an artificial technical reason why 2014/15 samples are high
Line 394: earlier in the paper you suggested that you discarded all but the most recent of samples from the same individual but here you mention that did in fact analyze them. It would be helpful to present these results rather than just mention them here. I missed them in the results section.
Line 516: ‘were identified to have’ could be replaced by ‘had’ for improved clarity.
Reviewer 3 Report
A brief summary
The paper describes as the authors said, the first study that examines from both a spatial and a temporal point of view the presence of phthalate metabolites in bottlenose dolphins. The paper presents a long-term (10-year) temporal period and a fine-scale spatial extent. The study includes the determination and screened for eight phthalate metabolites. Moreover, it uses a long-term photo-identification procedure that allows the identification of individuals. Aside from being the first study of phthalate metabolites in bottlenose dolphins, it is enforced by the long period of time including the fine-scale spatial extent as well as a robust statistical treatment of the data.
General concept comments
The manuscript is clearly structured, well written, sufficiently detailed, and easy to read. It includes an adequate number of references, not being detected lack of bibliography, or an abnormal number of self-citations. Although some of the references are not recent references it is comprehensive due to the lack of work in the field.
From the sampling strategy to the treatment of the data including the analytical schemes, the article is performed with an adequate scientific approach. Everything is clearly described (or correctly referenced) to be reproducible based on the details given in the appropriate sections. Besides the tables and figures effectively complete the text information. The data is treated under an intense statistical treatment, with the only deficit of more support about the need/utility of the use of the different statistics.
The conclusions of the article are consistent with the results of the study and the achievements obtained emphasize the importance of developing monitoring programs even for those contaminants that are already limited in their use or banned. The article identified areas more polluted and also a time contamination event (2014-2015). The article makes a correct evaluation of the obtained data and also puts forward the possibility of a future study that expands the sample timeframe to confirm whether acute contamination is likely to have occurred.
Round 2
Reviewer 1 Report
Thank you for the response, the edits look good!
Reviewer 2 Report
Thank you for addressing my queries and concerns